# A prospective observational study on persistent postoperative pediatric pain, 4P: The study protocol

**Johanna Broman**[1,2]*, **Niklas Nielsen**[1,2], **Anna K. M. Persson**[3,4]

**1** Department of Anesthesiology and Intensive Care, Helsingborg Lasarett, Helsingborg, Sweden,
**2** Department of Clinical Sciences Helsingborg, Lund University, Helsingborg, Sweden, **3** Department of Anesthesiology and Intensive Care Medicine, Halland Hospital Halmstad, Halmstad, Sweden, **4** Department of Clinical Sciences Malmö, Lund University, Malmö, Sweden

* johanna.broman@med.lu.se

**Data Availability Statement:** No datasets were generated or analysed during the current study. All relevant data from this study will be made available upon study completion.

## Abstract

### Background

Access to adequate pain treatment is a fundamental right, yet international data suggest that a considerable number of children experience acute and persistent pain. Little is known about the occurrence of both acute and persistent pain in children. The incidence of persistent postoperative pain in children is an unexplored area but international studies suggest that many children experience long-term pain after surgery, with a major impact on daily life. In 4P, Persistent Postoperative Pediatric Pain, we want to estimate the incidence of acute and persistent postoperative pediatric pain. The European Society for Paediatric Anaesthesiology has developed guidelines for perioperative pain management. We aim to examine how well these guidelines are followed and whether adherence to guidelines influences the pain experienced after surgery.

### Method

4P is a prospective observational study of children aged 1–17, planned for surgery in southern Sweden 2023–2024. After agreement from all caregivers, data concerning preoperative pain, pre-emptive analgesia, perioperative management and postoperative pain will be collected. Via an electronic management software, pain will be evaluated at home (or in hospital) at 24h, 3, 6 and 12 months after surgery. We will include 1000 patients.

### Discussion

4P will prospectively follow a large number of children after general pediatric surgeries and evaluate the occurrence of postoperative pain, both acute (APOP) and persistent (PPOP). The study will assess pain treatment regimens and identify risk factors associated with the development of acute and persistent pediatric postoperative pain.

### Trial registration

Prospectively posted at ClinicalTrials.gov, identifier NCT06035042.

**Funding:** JB has recieved grants from Stig and Ragna Gorthon Foundation Helsingborg AP has recieved grants from The Scientific council of Southern Sweden, Lund University, the Research and Educational fund in Halland County and Region Halland. https://www.gorthonstiftelsen.se https://www.vr.se/english.html https://www.lunduniversity.lu.se https://www.hh.se https://www.regionhalland.se The funders did not play any role in the study design, data collection, analysis, decision to publish or preparation of the manuscript.

**Competing interests:** The authors have declared that no competing interests exist.

## Introduction

Despite the fact that the Montreal Declaration from 2010 as well as the Swedish Convention on the Rights of the Child (Swedish law since January 1, 2020) establish that "access to adequate pain treatment is a fundamental right", it is estimated that 80% of the world's population is affected by inadequate pain management, and children are a particularly vulnerable group [1]. According to international data a considerable number of children (approximately 15 to 25%) experience persistent pain and this can, in addition to causing great personal suffering with non-attendance in school and social isolation, also entail a significant public health problem [2, 3]. 24–27% of children admitted to hospital suffer from moderate to severe pain, with the majority related to surgery [4, 5].

In adults, the prevalence of persistent postoperative pain (PPOP) ranges from 5 to 50% [3, 6] and in children it has been suggested to be around 20%, [6] but reliable data is missing [7]. There are a few studies suggesting the incidence of PPOP in children to be around 10–20%. In a retrospective study looking at children after cardiac surgery, with a median follow up of 3.8 years, 21% experienced persistent pain and in 10% the pain was moderate to severe (>4.0 NRS (Numeric Rating Scale, used to assess pain severity using a 0–10 scale, with zero meaning "no pain" and 10 meaning "the worst pain imaginable")) [8]. In a cross-sectional study, a structured telephone interview was carried out including children between the ages of two and 17 years who had undergone general, urologic or orthopedic surgery in the preceding ten months. Fifteen (13%) of 113 children reported current pain related to the surgery. The average intensity of pain was 4.2 on an NRS (0–10), and in most children the pain had some interference with daily activities [9]. Two studies on children at three and six months after orthopedic and general surgery showed, in 10.9% and 23% respectively, that the children experienced pain greater than 3 on the NRS [10, 11]. A meta-analysis from 2017 by Rabbitts et al. found only four studies reporting the prevalence of PPOP and with pain assessed at different time-points. They concluded the prevalence to be around 20% in mixed surgery [6] and summarized that further studies on this topic were urgently needed. Since 2017, we have found no prospective studies on the incidence of acute or persistent postoperative pain in children. Thus, the prevalence is uncertain.

It is known that children who develop long-term pain are at greater risk of suffering from pain-related problems than adults [6]. Surgery is a potential trigger for pain and is an important aspect to address [3]. A majority of adult patients undergoing surgery experience postoperative pain and inevitably surgery is sometimes necessary also in childhood. In addition, pain treatment after pediatric surgery is particularly challenging since each child is unique concerning age, development, and size and the optimal dose of analgesics can differ greatly from child to child. Results from studies in different countries show that pain treatment of children often is inadequate [12–14].

In adults the most important risk factors for postoperative pain, both acute and persistent, are age, female gender, pain before surgery [15], and psychosocial factors [16]. Concerning childhood pain, gender differences have been reported when it comes to headache and stomach pain, where girls tend to have more pain than boys [2]. Clinical studies on postoperative pain are consistent with this, suggesting that acute and persistent postoperative pain tend to be more prevalent in girls than boys [17]. Concerning age, some studies suggest that low age is associated with lower risk of PPOP [3, 12], yet other studies find that younger ages do not protect against PPOP after adjusting for confounding variables [11]. Larger studies are lacking. There are also implications of the influence of psychological factors. Children are highly dependent and influenced by the behavior of their caregivers. One potential risk factor is an increased level of parental anxiety related to their child's surgery [18]. Parental catastrophizing

scores have also been shown to influence, like parental mental health and chronic pain [7, 19]. Studies suggest that the parental influence on the pain response of the child increases as time from surgery passes [7].

The European Society for Paediatric Anaesthesiology (ESPA) has published guidelines for perioperative pain management in various common surgical procedures and interventions in children. These guidelines are in accordance with the WHO Pain Relief Ladder and created with the aim to improve perioperative analgesia for children in Europe [1]. Pain treatment guidelines are not always followed in clinical practice for various reasons. One example is that many pediatric clinics do not use regional anesthesia techniques to the same extent as in the adult population, even though it has been proven safe and effective [5].

The aim of the 4P study is to estimate the incidence of acute and persistent postoperative pain in children, identify risk factors for developing acute and persistent postoperative pediatric pain, and to investigate adherence to the European perioperative guidelines for pain management in children.

## Materials and methods

### General design

4P is a prospective observational study, designed in accordance with STROBE guidelines. Recruitment period of the study is Sept 1$^{st}$, 2023, to December 31$^{st}$, 2024 (Fig 1).

### Research ethics approval

This study was ethically approved by the the Swedish Ethics Review Authority on 1$^{st}$ of Dec 2022 with identification number: Dnr 2022–04821. The research protocol has also been pre-registered at clinicaltrials.gov under the record NCT06035042.

### Study setting

Children aged 1–17 years planned for surgery in southern Sweden (hospitals of Kristianstad, Trelleborg, Ystad, Malmö, Lund, Helsingborg, Ängelholm, Växjö, Halmstad, Gothenburg) will be included. This is a representation of hospitals of different sizes and settings, including university hospitals (Gothenburg, Lund and Malmö) with pediatric anesthesia units.

### Eligibility criteria

Inclusion criteria: age 1–17 years, planned for surgery in one of ESPA defined pediatric surgical procedures; inguinal hernia, circumcision/hypospadias/cryptorchidism, adenoidectomy/tonsillectomy/tonsillotomy, appendectomy, orthopedic acute fracture surgery.

Exclusion criteria: repeated surgery, inability of child or caregiver to speak/read Swedish, inability of child/caregiver to understand the implication of participation.

### Patient information and consent

Patients (15 years or older) or caregivers (if <15 years old) will be informed in writing a few days before surgery, and in person upon arrival to the hospital. Written and oral consent will be mandatory. In patients <15 years old we will seek acceptance from both caregivers in order to continue inclusion as demanded by the Swedish Ethical Review Authority.

### Study timeline

The timeline for the 4P-study protocol is depicted in Fig 2.

| | Study period | | | | | | |
|---|---|---|---|---|---|---|---|
| | **Enrolment** | | **T₀** | **T₁** | **T₂** | **T₃** | **T₄** |
| **Timepoint** | T -1-2 weeks | At home before surgery | Day of surgery | 24h | 3 months | 6 months | 1 year |
| **Enrolement** | | | | | | | |
| Eligibility screen | x | | | | | | |
| Informed consent | | x | | | | | |
| **Interventions:** | None – observational study | | | | | | |
| **Assessments** | | | | | | | |
| **Baseline variables:** | | | | | | | |
| Age | x | | | | | | |
| Gender | x | | | | | | |
| Preoperative pain evaluation | | x | | | | | |
| Sleep quality | | x | | | | | |
| Anxiety | | x | | | | | |
| Level of parental stress | | x | | | | | |
| **Outcome variables** | | | | | | | |
| APOP | | | x | x | | | |
| Adherence to anesthesia guidelines | | | x | x | x | x | x |
| PPOP | | | | | x | x | x |
| Influence of regional blocks | | | x | x | x | x | x |
| Influence of risk factors (age, gender, sleep quality, parental stress) | | | x | x | x | x | x |

**Fig 1. SPIRIT schedule.** Abbreviations: APOP = Acute postoperative pain: PPOP = Persistent postoperative pain.

## Measurement and definition of pain

Throughout this study, we will evaluate pain with different scales depending on age:

Age 1–3: FLACC (Face/Legs/Activity/Cry/Consolability Pain scale) Score range 0–10 [20]

Age 4–14: FPS-Revised (The Faces Pain Scale–Revised) Score range 0–10 [21]

Age 15–18: NRS (Numeric rating scale) Score range 0–10 [22]

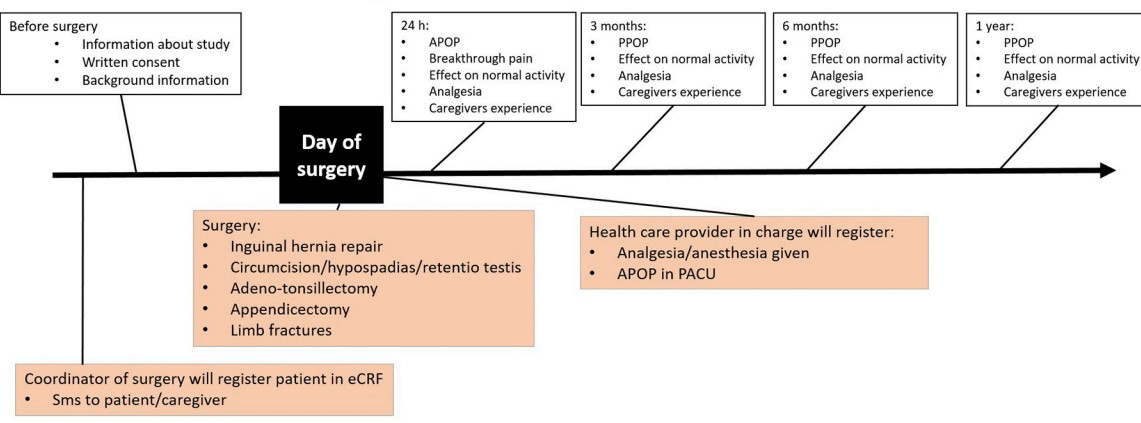

**Fig 2. Flow-chart depicting timeline.** Abbreviations: eCRF = electronic case report form; APOP = acute postoperative pain; PPOP = persistent postoperative pain; PACU = post anesthesia care unit.

Higher values indicate more pain, with zero meaning "no pain".

In statistical analyses we will compare levels (0–10) independent of scale.

We will evaluate preoperative pain via a questionnaire sent to the patient prior to surgery and ask if the child has experienced recurrent pain before surgery (yes/no) and level of pain (using FLACC, FPS or NRS (0–10)), and if yes, for how long and whether or not this has influenced normal activities. We will also ask if the child has experienced pain 24 hours prior to surgery (yes/no) and note whether analgesics were offered.

We will evaluate acute postoperative pain in the postoperative care unit (PACU), where pain will be measured repeatedly by nursing staff using FLACC, FPS and NRS (0–10). The worst pain measured during the visit will be registered and used for analysis. For secondary outcomes a cut-off for moderate- to severe acute postoperative pain, APOP, is set to 4.0 (FLACC, FPS or NRS).

Twenty-four hours after surgery, we will evaluate pain with a questionnaire where the patient/caregiver is asked to note the worst pain experienced since they left the PACU (FLACC, FPS, NRS (0–10)).

We will evaluate persistent postoperative pain at three months, six months and one year post surgery with a questionnaire where patients/caregivers are asked to note present pain (FLACC, FPS, NRS (0–10)). PPOP is set to 1 (FLACC, FPS or NRS) or higher and moderate to severe PPOP is defined as pain $\geq 4$ [6, 8] but the existence will be accompanied with a description of level.

### Evaluation of potential risk factors

We will register gender and age, allowing for a comparison of groups.

We will evaluate sleep quality and anxiety via a questionnaire sent to the patient prior to surgery, asking if the child experienced problems before surgery, and if yes, for how long and whether or not it influenced normal activities.

We will register the level of parental stress (1–10) at 24 hours, three and six months and one year after surgery.

## Study outcomes

The primary outcome is the incidence of acute postoperative pain in children.

Secondary outcomes are adherence to guidelines from ESPA regarding perioperative analgesia, incidence of persistent postoperative pain, the influence of regional blocks on APOP and PPOP, the influence of parental stress, age and gender on APOP and PPOP, and the association of the choice of pain treatment (premedication/long vs short acting opioid/paracetamol/ NSAID/ketamine/local anesthetics/betamethasone) on APOP and PPOP (Fig 1).

## Sample size

We will screen all patients meeting the inclusion criteria at the participating hospitals between September 2023 –December 2024 for study eligibility.

The primary outcome in 4P is descriptive and hypothesis generating, therefore we have elected not to perform a power calculation. Previous studies [6] give us reason to believe that 20% of the patients will experience PPOP and possibly APOP and thus 1000 children included will allow for analysis of risk factors.

## Data management

We will use the data management software Entermedic (Entergate ®, Halmstad Sweden) where patient data and contact information are stored. We will keep an electronic chart for each patient included in the platform Entermedic. The primary investigator will register information regarding drugs used during anesthesia in Entermedic, as well as APOP in the PACU. The patient/caregiver will register acute postoperative pain between 1.5 and 24 hours after surgery, and PPOP after three and six months and one year in the eCRF (Entermedic). We have programmed the software to send out reminder text messages with a narrow interval, avoiding inaccurate data. When all data points are complete, the system stops sending out text reminders. As a back-up, investigators check for potential incomplete data regularly.

Entermedic is specifically designed for simplicity and security regarding data management. Ethical approval from the Swedish Ethical Review Authority is mandatory when using the service. Entergate® has their servers in Sweden and do not outsource data for any reason. Entermedic is also certified with the international standard regarding security standard (ISO 27000:1) and completely adapted to the European Union's demands on General Data Protection Regulation (GDPR). The service is also adapted to fit Swedish law for patient data reporting, holding and management (Department of social affairs number 2008:335, 2010:659 and 1985:562). Furthermore, Entermedic is also approved by the Swedish Medical Products Agency.

## Status of study

We included the first patient Sept 1st, 2023. By Aug. 1st 2024 we have 959 registered eligible and 380 included patients.

## Statistical methods

We will present the incidence of APOP and PPOP at different timepoints as percent with confidence intervals in tables. We will present the distribution of age, gender and type of surgery in tables. We will also present mean APOP and PPOP in different age-groups. The time frame is acute—within 24 hours (APOP) and at three and six months and one year after surgery (PPOP).

Adherence to ESPA guidelines: We will present how well the guidelines are followed in a qualitative description using percentages and confidence intervals. We will compare the groups (Yes: guidelines followed/No: guidelines not followed) concerning APOP and PPOP with Mann Whitney, U-test and description of the deviation. The time frame is perioperative, within 24 hours (APOP) and at three and six months and one year after surgery (PPOP).

Influence of regional blocks on APOP and PPOP: we will compare groups with Mann Whitney, U-test. The time frame is acute–within 24 hours (APOP) and at three months after surgery (PPOP).

We will evaluate risk factor analysis regarding parental stress, age and gender with logistic regression. The time frame is acute—within 24 hours (APOP) and at three and six months and one year after surgery (PPOP). To investigate whether children who experience APOP ($\geq$4.0) have increased risk of PPOP (3 months) we will use logistic regression.

We will investigate the effect of preemptive analgesia, choice of perioperative opioid given during TIVA (Alfentanil, Remifentanil), and method of anesthesia (volatile compared to total intravenous) on APOP. Group comparison of APOP in PACU and at 24 h with Mann Whitney.

## Monitoring

This study is strictly observational and hence has no appointed Data Monitoring Committee (DMC) and no interim analysis are planned.

## Protocol amendments

New approval will be sought out for any amendments (eligibility criteria, allocation, intervention protocol).

## Consent or assent

Participation will be voluntary, and consent can be rejected at any time. Informed consent from each participating patient (aged 15 or older) or one or both caregivers (age 1–14) will be signed and registered in Entermedic.

## Dissemination policy

The trial results will be communicated to participants, healthcare professionals and the public via publication in medical journals. There are no planned publication restrictions.

## Discussion

There are a variety of strengths in our study. 4P is a multicenter study, with different characteristics of the participating hospitals, providing a broad picture of the pediatric anesthesia being conducted in southern Sweden. All registration forms are filled out electronically directly following patient contact or by the patient or caregiver directly, hence allowing an accurate and current observation. We use pain scales adjusted for age, all on a numerical scale 1–10 which allows for comparison of groups.

At the same time, it should be noted that several limitations may exist. Persistent postoperative pain (PPOP) is defined as pain lasting more than three months after surgery. The actual value used for defining existing PPOP differs in different studies, making comparison of results difficult [3, 8, 9, 11, 23] and no international consensus has yet been met [24]. We have chosen to use any pain after 3 months, in other words pain level $\geq$ 1 with the respective scale depending on age of the child but also to describe the level at the respective timepoint.

The centers involved are diverse, which may lead to a bias when it comes to inclusion and type of surgery. All hospitals anesthetize children from the age of one. At the more rural sites the primary surgery type will be ear nose throat surgery, and we suspect that they will represent a large bulk of the patients. This is also a type of surgery where we have widespread national guidelines on perioperative treatment [25, 26]. The University hospitals (Gothenburg, Lund and Malmö) handle the younger children, as well as the most critically ill. Some types of surgery will primarily be performed at these centers, namely inguinal hernia, circumcision/hypospadias/cryptorchidism. Appendectomies and fracture surgery will be performed at all sites.

According to the Swedish Ethics Review Authority it is mandatory for both caregivers, if there are more than one, to leave written consent in advance. This makes the process of inclusion time-consuming, and we suspect we might fail to include a proportion of patients due to this, thus risking confounding due to many missed to include.

In summary, with this prospective observational study we aim to estimate the incidence of acute and persistent postoperative pain in a pediatric population in Sweden, illustrate what pain treatment is commonly used and hopefully identify factors possible to improve. This important topic is poorly covered and there is a significant knowledge gap.

## Acknowledgments

To everyone involved in 4P at the hospitals of Kristianstad, Trelleborg, Ystad, Malmö, Lund, Helsingborg, Ängelholm, Växjö, Halmstad, Göteborg. Thank you for your enthusiasm, time and engagement. Thank you, Stefan Lundeberg, for your support and inspiration.

## Author Contributions

**Conceptualization:** Johanna Broman, Niklas Nielsen.

**Data curation:** Johanna Broman, Anna K. M. Persson.

**Formal analysis:** Johanna Broman, Anna K. M. Persson.

**Funding acquisition:** Johanna Broman, Anna K. M. Persson.

**Investigation:** Niklas Nielsen.

**Methodology:** Johanna Broman, Niklas Nielsen, Anna K. M. Persson.

**Project administration:** Johanna Broman, Anna K. M. Persson.

**Supervision:** Niklas Nielsen, Anna K. M. Persson.

**Validation:** Niklas Nielsen.

**Visualization:** Niklas Nielsen.

**Writing – original draft:** Johanna Broman, Anna K. M. Persson.

**Writing – review & editing:** Johanna Broman, Niklas Nielsen, Anna K. M. Persson.

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
