## [Decision Letter · Decision Letter 0]

22 Nov 2024

PONE-D-24-39056A prospective observational study on Persistent Postoperative Pediatric Pain, 4P: The study protocolPLOS ONE

Dear Dr. Broman, 

Thank you for submitting your manuscript to PLOS ONE. After careful consideration, we feel that it has merit but does not fully meet PLOS ONE’s publication criteria as it currently stands. Therefore, we invite you to submit a revised version of the manuscript that addresses the points raised during the review process.

Protocols.io assigns your protocol its own identifier (DOI) so that it can be cited independently in the future. For instructions see: https://journals.plos.org/plosone/s/submission-guidelines#loc-laboratory-protocols. Additionally, PLOS ONE offers an option for publishing peer-reviewed Lab Protocol articles, which describe protocols hosted on protocols.io. Read more information on sharing protocols at https://plos.org/protocols?utm_medium=editorial-email&utm_source=authorletters&utm_campaign=protocols.

We look forward to receiving your revised manuscript.

Kind regards,

Frances Chung, MBBS, MD,  FRCPC

Academic Editor

PLOS ONE

Journal Requirements:

3. Please note that funding information should not appear in the Acknowledgments section or other areas of your manuscript. We will only publish funding information present in the Funding Statement section of the online submission form. Please remove any funding-related text from the manuscript. 

Reviewers' comments:

Reviewer's Responses to Questions

**Comments to the Author**

1. Does the manuscript provide a valid rationale for the proposed study, with clearly identified and justified research questions?

Reviewer #1: Yes

2. Is the protocol technically sound and planned in a manner that will lead to a meaningful outcome and allow testing the stated hypotheses?

Reviewer #1: Yes

3. Is the methodology feasible and described in sufficient detail to allow the work to be replicable?

Reviewer #1: Yes

4. Have the authors described where all data underlying the findings will be made available when the study is complete?

Reviewer #1: Yes

5. Is the manuscript presented in an intelligible fashion and written in standard English?

Reviewer #1: Yes

6. Review Comments to the Author

You may also provide optional suggestions and comments to authors that they might find helpful in planning their study.

Reviewer #1: Although the study has been performed before numerous times, I salute you for tackling this important subject.

I would add that many variables should taken into consideration in your statistical model as surgery time, surgical complications, patient comorbidities and psychological evaluation of the child before surgery

7. **Do you want your identity to be public for this peer review?** 

Reviewer #1: **Yes: **Mahmoud Elfiky

---

## [Author Response · Author response to Decision Letter 0]

10 Dec 2024

Thank you for considering this protocol for publication and for your comments. Below you can find our responses to comments from the editor and reviewer: 

Journal Requirements: 

Author's comment: We have adapted the manuscript according to the instructions. 

Author´s comment: The grants we have received do not have grant numbers. We have corrected the information given under “funding information” and “financial disclosure” and deleted the information on funding from the manuscript file. 

3. Please note that funding information should not appear in the Acknowledgments section or other areas of your manuscript. We will only publish funding information present in the Funding Statement section of the online submission form. Please remove any funding-related text from the manuscript. 

Author´s comment: Thank you, it has now been removed from the manuscript. 

Author´s comment: We have provided captions at the end of the manuscript accordingly. 

Author´s comment: We cannot find any retracted manuscripts. Please be more specific if there is something we have missed. 

Reviewers' comments:

Reviewer's Responses to Questions

Comments to the Author

1. Does the manuscript provide a valid rationale for the proposed study, with clearly identified and justified research questions?

Reviewer #1: Yes

2. Is the protocol technically sound and planned in a manner that will lead to a meaningful outcome and allow testing the stated hypotheses?

Reviewer #1: Yes

3. Is the methodology feasible and described in sufficient detail to allow the work to be replicable?

Reviewer #1: Yes

4. Have the authors described where all data underlying the findings will be made available when the study is complete?

Reviewer #1: Yes

5. Is the manuscript presented in an intelligible fashion and written in standard English?

Reviewer #1: Yes

6. Review Comments to the Author

Reviewer #1: Although the study has been performed before numerous times, I salute you for tackling this important subject.

I would add that many variables should taken into consideration in your statistical model as surgery time, surgical complications, patient comorbidities and psychological evaluation of the child before surgery

Author’s comment: Thank you for your important comments. 

We are aware of the fact that several variables need to be taken into consideration. 

Surgery time, surgical complications: we have not included this in the registration and these important issues need to be addressed in the discussion section of future publications. We have grouped our types of surgery into groups of similar surgery types as well as expected similar surgery times. It would have been of interest to know the exact surgery time and we will address this in discussions of future publications.

Patient comorbidities and psychological evaluation before surgery: We do not register comorbidities; however, we register both pain and psychological well-being before surgery. 

7. Do you want your identity to be public for this peer review? 

Reviewer #1: Yes: Mahmoud Elfiky

While revising your submission, please upload your figure files to the Preflight Analysis and Conversion Engine (PACE) digital diagnostic tool, https://pacev2.apexcovantage.com/. 

Author’s comment: Figure 1 and 2 have been uploaded in PACE. The revised version will be added in the revised version of the submission.

---

## [Editor Report · Decision Letter 1]

13 Dec 2024

A prospective observational study on Persistent Postoperative Pediatric Pain, 4P: The study protocol

PONE-D-24-39056R1

Dear Dr. Broman,

We’re pleased to inform you that your manuscript has been judged scientifically suitable for publication and will be formally accepted for publication once it meets all outstanding technical requirements.

Kind regards,

Frances Chung, M.B.B.S, F.R.C.P.C

Academic Editor

PLOS ONE
---

## [Editor Report · Acceptance letter]

26 Dec 2024

PONE-D-24-39056R1 

PLOS ONE

Dear Dr. Broman, 

I'm pleased to inform you that your manuscript has been deemed suitable for publication in PLOS ONE. Congratulations! Your manuscript is now being handed over to our production team.

Kind regards, 

on behalf of

Dr. Frances Chung 

Academic Editor

PLOS ONE